



# Contributions of the synoptic meteorology to the seasonal CCN cycle over the Southern Ocean

Tahereh Alinejadtabrizi[1,2,3], Yi Huang[3,4], Francisco Lang[1,5], Steven Siems[1,2], Michael Manton[1], Luis Ackermann[6], Melita Keywood[7,8], Ruhi Humphries[7,8], Paul Krummel[7], Alastair Williams[9], and Greg Ayers[10]

[1]School of Earth, Atmosphere and Environment, Monash University, Melbourne, Victoria, Australia
[2]Australian Research Council Securing Antarctica's Environmental Future (SAEF), Melbourne, Victoria, Australia
[3]Australian Research Council Centre of Excellence for Climate Extremes (CLEX), Melbourne, Victoria, Australia
[4]School of Geography, Earth and Atmospheric Sciences, The University of Melbourne, Melbourne, Victoria, Australia
[5]Department of Geophysics, Universidad de Concepción, Concepción, Chile
[6]Australian Bureau of Meteorology, Melbourne, Victoria, Australia
[7]CSIRO Environment, Melbourne, Victoria, Australia
[8]Australian Antarctic Program Partnership, Institute for Marine and Antarctic Studies, University of Tasmania, Hobart, Tasmania, Australia
[9]Environmental Research, ANSTO, Lucas Heights, New South Wales, Australia
[10]Visiting Scientist Emeritus, Bureau of Meteorology, Melbourne, Victoria, Australia

**Correspondence:** Tahereh Alinejadtabrizi (tahereh.alinejadtabrizi@monash.edu)

**Abstract.** Cloud Condensation Nuclei (CCN) play a fundamental role in determining the microphysical properties of low-level clouds, crucial for defining the energy budget over the Southern Ocean (SO), a region dominated by low-level clouds. Despite this importance, many aspects of the CCN budget over the SO remains poorly understood including the role of the synoptic meteorology. In this study, we classify the dominant synoptic meteorology over kennaook/Cape Grim Observatory (CGO) and

5    examine its influence on the seasonal variation of the CCN concentration ($N_{CCN}$).

Our analysis identifies six distinct synoptic regimes: three prevalent in the austral winter, when the subtropical ridge (STR) is strong and centred at lower latitudes, and three in the austral summer, when the STR shifts to higher latitudes. Distinct winter and summer 'baseline' regimes contribute to the seasonal cycle in $N_{CCN}$ over the SO with the winter baseline regime characterised by heavier precipitation, a deeper boundary layer and lower $N_{CCN}$. An analysis of air mass back trajectories,

10   specifically at the free troposphere level, supports this distinction, with wintertime air masses originating over higher latitudes. Across these two baseline regimes we observe a significant inverse relationship between precipitation and $N_{CCN}$, underscoring the role of precipitation in reducing $N_{CCN}$ over the SO.

Using forward trajectories within this synoptic framework, we examine the transport of continental air masses over the SO, finding that frontal air masses more frequently reach high latitudes during winter. We conclude that the location of the STR

15   can moderate the advection of air masses between Antarctica and kennaook/Cape Grim.



## 1 Introduction

Low-altitude clouds, frequently found in or near the marine atmospheric boundary layer (MABL), are prevalent over mid-latitude oceans (Wood, 2012) and are a pivotal component of the Earth's climate system (Tselioudis et al., 2021) due to their direct impact on both the energy budget and hydrological cycle (Trenberth and Fasullo, 2010; Williams et al., 2013; Bodas-Salcedo et al., 2014; Tan et al., 2016; Schuddeboom and McDonald, 2021). These clouds are not only notoriously difficult to simulate accurately within climate models (Forbes and Ahlgrimm, 2014; Kay et al., 2016) but also exhibit a profound impact on climate sensitivity of these simulations, especially over the Southern Ocean (SO), as highlighted in the latest phase of the Coupled Model Intercomparison Project (CMIP6) (Zelinka et al., 2020). The radiative properties of these clouds are highly sensitive to both their macrophysics and microphysics (Wood, 2012; Wood et al., 2012), such as cloud droplet number concentration ($N_d$). Evidence from the Southern Ocean Clouds, Radiation, Aerosol Transport Experimental Study (SOCRATES) emphasized the intimate connection between $N_d$, cloud condensation nuclei concentrations ($N_{CCN}$), and aerosol properties in this region. Such insights highlight the critical role of aerosols, particularly cloud condensation nuclei (CCN), in shaping cloud properties and radiative effects over the SO (McFarquhar et al., 2021) and the complex interplay between aerosols, cloud formation, precipitation and the local dynamics and thermodynamics of the MABL.

The $N_{CCN}$ over the SO has been studied for decades (e.g., Gras, 1990, 1995; Ayers et al., 1997; Gras and Keywood, 2017; Humphries et al., 2021) due to its importance and the availability of long-term, high-quality field observations. Located at the northwest tip of Tasmania (40°41'S, 144°41'E), the kennaook/Cape Grim Baseline Air Pollution Station (CGO) has been providing unique access to pristine air masses off the SO during 'baseline' conditions (Gras and Keywood, 2017; Humphries et al., 2023) since 1976. This programme is the principal Australian contribution to the World Meteorological Organization (WMO) Global Atmosphere Watch (GAW) (Gras and Keywood, 2017). From the earliest observations, the CGO record has revealed a robust seasonal cycle in CCN (Bigg et al., 1984; Ayers et al., 1997; Gras and Keywood, 2017; Humphries et al., 2023). During the austral winter (JJA), the $N_{CCN}$ is at a minimum while peaks are observed over the summer months (DJF).

Dimethylsulphide (DMS), primarily originating from planktonic algae in seawater, emerges as a substantial source of CCN over oceanic regions (Charlson et al., 1987). While marine biological sources predominantly govern $N_{CCN}$ during the summer months, multiple elements contribute to CCN levels throughout the year over the SO (e.g., Ayers and Cainey, 2007; Korhonen et al., 2008; Quinn and Bates, 2011; Twohy et al., 2021). Coarse mode sea salt, generated by wind-driven processes, play a crucial role in CCN composition year-round (Hudson et al., 2011; Quinn et al., 2014; Sanchez et al., 2018). Beyond these primary contributors, various other sinks and sources influence the CCN budget over the SO (e.g., Raes, 1995; Bates et al., 1998; Vallina et al., 2006; Sanchez et al., 2021) that have not been so extensively studied.

Early simulations of the CCN budget within the SO MABL were driven by the CGO record (Ayers et al., 1995), demonstrating the importance of the seasonality of the biogenic activity within the surface fluxes. Such simulations were arguably limited, as a complete 1-D CCN budget of the MABL not only needs surface sources but must also include entrainment from the free troposphere as a potential source (Clarke et al., 1998; Capaldo et al., 1999; Katoshevski et al., 1999; Wood et al., 2012; Rose



et al., 2017). Since new particle formation is rare in the MABL, the exchange with the free troposphere can supply particles that
grow into CCN (Zheng et al., 2018, 2021). This contribution depends on condensable gases, temperature, humidity, and sink
availability. Further, such simple budget models need to include the sink terms from coalescence scavenging and wet deposition
(Feingold et al., 1996; Mechem et al., 2006; Wood, 2006; Kang et al., 2022; Alinejadtabrizi et al., 2024), although no such
observations have routinely been available. Kang et al. (2022), employing the SOCRATES campaign over the SO along with
a simplified but more comprehensive budget model (developed initially by Wood et al. (2012)), highlighted entrainment from
the free troposphere as a crucial source during the summertime and coalescence scavenging as a key sink of CCN over the SO
(Sanchez et al., 2021).

Alinejadtabrizi et al. (2024) further correlated the observed seasonality in CCN with cloud morphology, precipitation and
their meteorological controls (specifically atmospheric stability) observed upwind of CGO. Specifically, when open Mesoscale
Cellular Convection (MCC), a MABL cloud which is more common over the SO during the winter (Danker et al., 2022; Lang
et al., 2022, 2024), was present heavier and more frequent precipitation events were recorded coinciding with lower $N_{CCN}$.
This underscores the efficiency of wet deposition in removing CCN from MABL. This study further emphasized the need to
consider broader meteorological influences, when investigating the seasonality of $N_{CCN}$ and its implications for aerosol-cloud
interactions in the pristine SO environment.

Located at 41°S, the seasonal cycle of the meteorology governing CGO, reflects the annual advance and retreat of the Hadley
cell and subtropical ridge (STR) (e.g., Pittock, 1973; Dima and Wallace, 2003; Larsen and Nicholls, 2009; Cai et al., 2011).
Defined by the mean latitude and intensity of high-pressure systems near the midlatitudes, the STR is highly correlated to
both seasonal rainfall and wind patterns (Larsen and Nicholls, 2009; Grose et al., 2015) and temperatures (Pepler et al., 2018)
across Australia. Mace and Avey (2017) documented a seasonal cycle in the meteorology, specifically cloud properties and
precipitation processes in warm clouds, over the SO using the A-Train satellite data (consistent with other works e.g., Boers
et al. (1998); McCoy et al. (2014, 2015); Huang et al. (2015); Fletcher et al. (2016); Lang et al. (2018, 2022, 2024)).

Moving beyond the biogenic production of DMS, our investigation aims to extend our understanding of the role of the synoptic
meteorology in shaping the observed seasonal cycle in the $N_{CCN}$ over CGO under baseline conditions. Specifically, we seek to
better appreciate the role of the synoptic meteorology in defining both the seasonal precipitation and free troposphere transport
of aerosols. Employing a K-means clustering algorithm, we first define the synoptic meteorology over CGO, which includes
separate clusters for wintertime and summertime baseline conditions. Observations of precipitation underscore the significant
role of wet deposition as a sink term contributing to the observed seasonality. Using back trajectories for these synoptic clusters,
we also examine the seasonality in free troposphere transport across the SO. Using radon observations as a proxy for terrestrial
influences, we find further evidence of meteorological controls in defining the CCN budget. Finally, from this framework we
examine the seasonality of the free troposphere transport across the greater extent of the SO.



## 2  Data and Methodology

The meteorological data set is taken from the fifth generation of European ReAnalysis (ERA5), produced by the European Centre for Medium-Range Weather Forecasts (ECMWF) (Hersbach et al., 2020) which is available through the Copernicus Climate Change Service Climate Data Store (https://cds.climate.copernicus.eu). Our analysis employs 8036 virtual soundings taken twice per day (00:00 and 12:00 UTC) over a period of 11 years (2011-2021) over the grid point nearest to CGO.

A simple K-means clustering algorithm (Anderberg, 1973) is employed to classify the 11 years of synoptic meteorology based on the low-altitude thermodynamic structure. K-means clustering algorithm has been widely utilized over the SO to investigate cloud regimes (e.g., Gordon and Norris, 2010; Haynes et al., 2011; Mason et al., 2014), the climatology of MABL (Truong et al., 2020, 2022) and MABL's responses to synoptic forcing (e.g., Hande et al., 2012; Lang et al., 2018; Montoya Duque et al., 2022, 2023). Consistent with the approach of Lang et al. (2018); Truong et al. (2020), a set of 15 variables is employed for the clustering analysis. These include four variables (the temperature, relative humidity, zonal and meridional winds) at the three different atmospheric levels (925, 850, and 700 hPa) and three surface variables (pressure, air temperature and relative humidity). Standardization is applied to each variable before clustering. Initially, the analysis considers the number of the clusters (K) ranging from 2 to 10 (results not shown). Ultimately, 6 clusters were chosen as it represents the minimum number of clusters that effectively differentiates the synoptic meteorology.

Hourly observations of CCN spanning eleven years are available from the CGO, located at the northwest tip of Tasmania. The particle measurements at CGO, initiated in the mid-1970s with a range of technologies, align with the recommendations of the World Meteorological Organisation's Global Atmosphere Watch (WMO-GAW) programme under the Aerosol Programme (Gras and Keywood, 2017; Humphries et al., 2023). This study utilizes the $N_{CCN}$ at 0.5% supersaturation (Model CCN-100, Droplet Measurement Technologies, Longmont, CO, USA) for the same twice-daily 8036 soundings. CCN data for other supersaturation levels are not available at these times. Data can be accessed through the World Data Centre for Aerosols (http://www.gaw-wdca.org/). Additionally, hourly measurements of radon, as an unequivocal tracer of terrestrial influences on sampled air masses (Zahorowski et al., 2013; Chambers et al., 2015, 2018), are conducted using a dual-flow-loop two-filter atmospheric radon detector over the CGO station (Whittlestone and Zahorowski, 1998; Williams and Chambers, 2016). The hourly precipitation data, in $\mathrm{mm\,hour^{-1}}$, were also obtained from the Australian Bureau of Meteorology rain gauge stationed close to CGO (Station ID: 091331) for the corresponding times.

Traditionally for CGO, the baseline sector is defined as periods with surface wind directions between 190° and 280° (Ayers and Gillett, 2000; Gras et al., 2009), coupled with radon concentrations below various thresholds such as $150\,\mathrm{mBq\,m^{-3}}$ (Gras and Keywood, 2017). No distinction is made for season. We define this constraint as "CGO baseline" hereafter, as opposed to the "Winter baseline" and "Summer baseline" clusters produced from our cluster analysis. Air sampled in the CGO Baseline sector has typically traversed several thousand kilometres across the SO, minimizing recent anthropogenic and terrestrial influences (Ayers and Gillett, 2000; Gras and Keywood, 2017).



The Hybrid Single Particle Lagrangian Integrated Trajectory (HYSPLIT) model was employed for running the forward/back trajectories (Draxler and Hess, 1998), to analyze the source and transportation patterns of the air parcels. For the back trajectories, hourly ERA5 data served as the input for meteorological parameters. However, the forward trajectories using ERA5 encountered a limitation where they ceased running upon reaching 180 degrees east. To address this issue, we opted to utilize the Global Data Assimilation System (GDAS) data, which are available in daily format with a 0.5°horizontal resolution, ensuring comprehensive coverage and continuity throughout the forward trajectory simulations. A comparison between ERA5 and GDAS up to the 180°E limit showed no significant differences in the back trajectories, indicating that GDAS is a reliable alternative for our forward trajectory analysis.

## 3   Synoptic classification

The application of the K-means clustering algorithm (K = 6) to the 11 years of atmospheric profiles (twice per day) over CGO has revealed strong seasonality in the synoptic meteorology (Figure 1): three clusters, shown in blue, have higher frequencies during the austral winter months (JJA) and September, while the remaining three, shown in red, are more common in the austral summer months (DJF) and March. To simplify interpretation, these clusters are first split into two main groups accordingly: winter clusters, identified with their names beginning with 'W' and summer clusters, denoted with names beginning with 'S'. The clusters are also named based on their specific synoptic meteorological characteristics, which will be discussed later in this section. Throughout all plots, the winter clusters consistently appear in blue and the summer clusters in red. It should be noted that due to the inherent variability of the synoptic meteorology, some data points from winter may be classified within the summer clusters and vice versa.





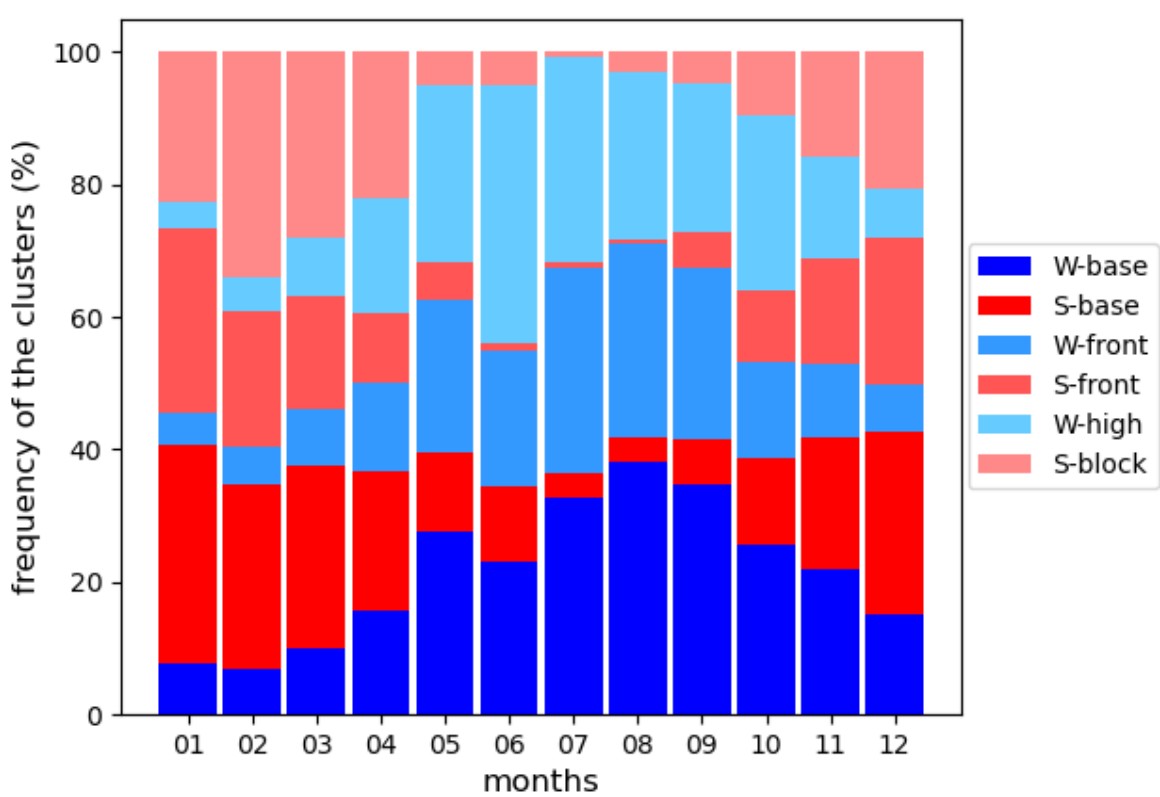

**Figure 1.** Monthly variation in the frequency of 6 clusters (2011-2021).

The analysis of the location of the STR and its intensity has been done using the mean sea level pressure (MSLP) data from ERA5 spanning eleven years (two times per day). Zonal MSLP values were calculated for each latitude ranging from 10-60°S within the longitudinal span of 110-160°E for every month. Then the maximum MSLP (intensity) and the latitude at which it occurred were determined and plotted (Figure 2).



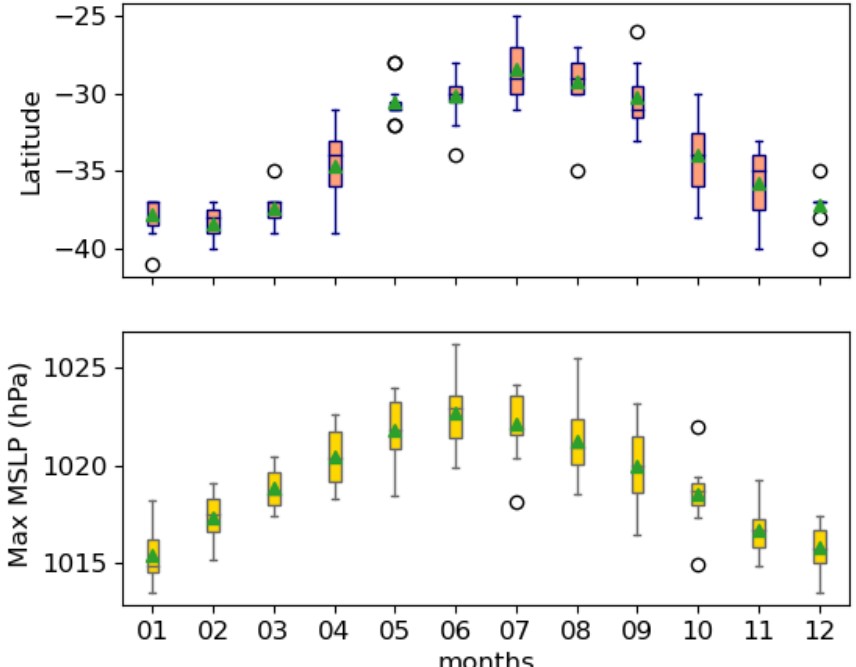

**Figure 2.** Box-whisker plots showing the latitude where the maximum MSLP occurs (top) and the maximum MSLP (intensity) itself (bottom) (2011-2021). The green triangles represent the mean values, the middle line inside the boxes represents the median and the top and bottom of the boxes indicate the 25th and 75th percentiles.

The seasonal cycles of all six clusters are seen to be highly correlated with the migration of the STR, reflecting the role of
the synoptic meteorology in determining the air mass being observed at CGO. The top plot in figure 2 illustrates the well-documented annual progression of the STR for our 11-year period, migrating to lower latitudes in austral winter (JJA) and higher latitudes in austral summer (DJF), which is consistent with the literature (e.g., Williams and Stone, 2009; Larsen and Nicholls, 2009). The bottom plot in figure 2 reveals that as the STR shifts equatorward, its maximum pressure increases, whereas summertime pressures are lower when the STR is located further poleward, again consistent with the findings of
Larsen and Nicholls (2009).

The MSLP composite plots (Figure 3) illustrate the seasonal migration of the STR with the high-pressure centres being located over lower latitudes in the winter clusters (top row) and over higher latitudes in the summer clusters (bottom row). Moving beyond the seasonality, the composite MSLP plots also reveal intraseasonal differences between the clusters, which is further used for synoptic classification. Starting with the left-hand column (Figure 3a and d), both plots have an MSLP contour aligned
from southwest to northeast in the immediate vicinity of CGO (highlighted by star sign in Figure 3), which suggests that the station is observing a south-westerly, or baseline air mass on average. Moving to the middle column (Figure 3b and e), the MSLP contours have the opposite slope (northwest to southeast) in the vicinity of CGO suggesting a continental influence to





the air mass. The remaining two clusters (Figure 3c and f), i.e., the right-hand column, both have CGO located close to the high-pressure centres, suggesting weak surface winds and a strong inversion.

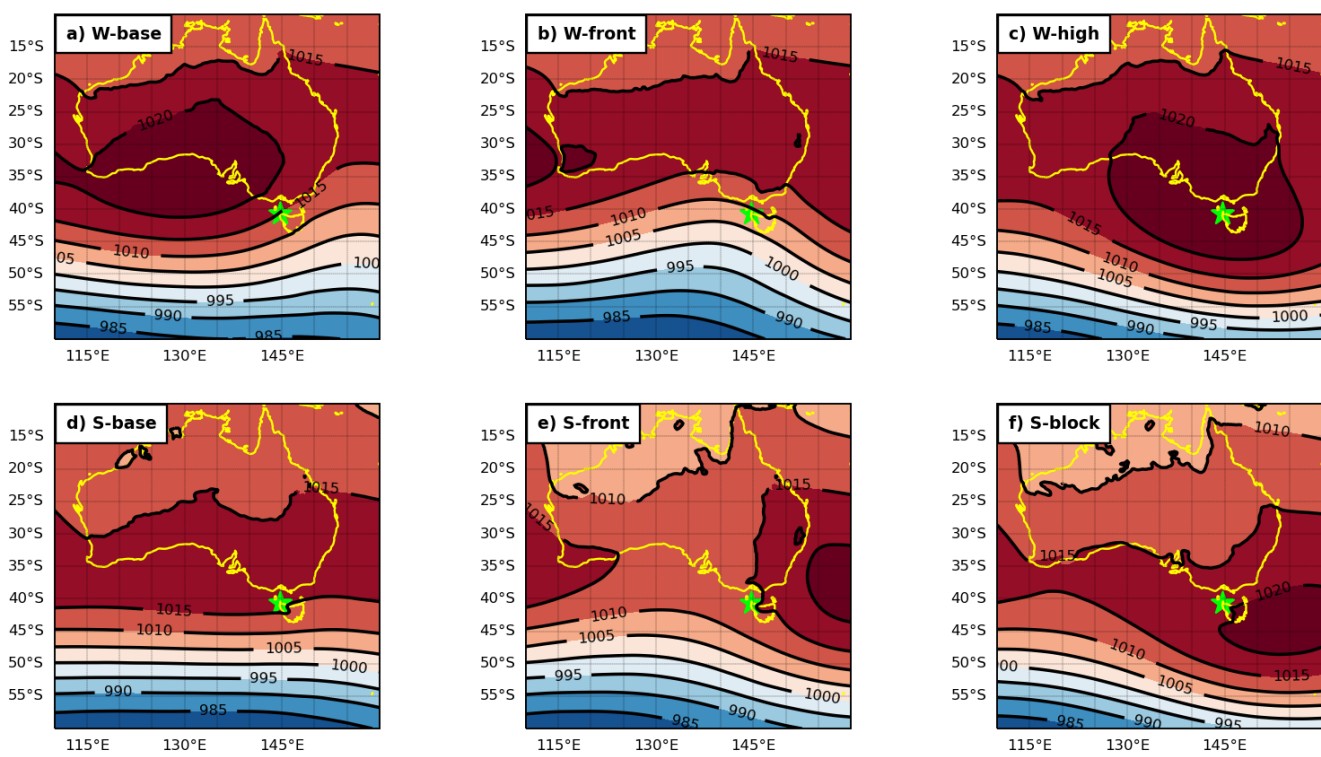

**Figure 3.** MSLP composite for six clusters (2011-2021). Winter clusters are displayed in the top row while the bottom row shows the summer clusters. The star symbol highlights the location of CGO.

Next, we examine composite soundings for the six clusters using the ERA5 datasets (Figure 4). Starting with the left-hand column again (Figure 4a and d), a south westerly wind is observed at the surface, consistent with the composite MSLP plots (Figure 3a and d). The wintertime composite (Figure 4a) has a more southerly heading of the two. The heading largely remains unchanged moving up out of the boundary layer into the free troposphere, although it turns more westerly for the summertime cluster (Figure 4d). The inversion is seen to be deeper in the winter, and the relative humidity through the boundary layer suggests precipitation from boundary layer clouds may be relatively frequent. Conversely, the inversion is shallower (~900 hPa) for the summer composite (Figure 4d). Again, the 1000 hPa winds align closely with the definition of CGO baseline conditions (e.g., Ayers et al., 1995; Gras and Keywood, 2017; Humphries et al., 2023), prompting us to designate these as summertime (S-base) and wintertime (W-base) baseline clusters.

Moving to the middle column (Figure 4b and e), a strong north westerly wind is evident through the free troposphere leading us to label them as frontal clusters, W-front and S-front, respectively. Amongst all six clusters, the relative humidity is greatest for W-front suggesting heavier precipitation rate, on average. The S-front cluster shows strong turning through the boundary





layer and is very dry through the free troposphere. Turning attention to the last two clusters in the right-hand column (Figure 4c and f), the top composite, displays zero wind speed consistent with the MSLP composite (Figure 3c) earning it the label of the high-pressure cluster (W-high). The bottom one, on the other hand, illustrates a blocking system commonly observed

during the summer season (S-block) (Risbey et al., 2013).

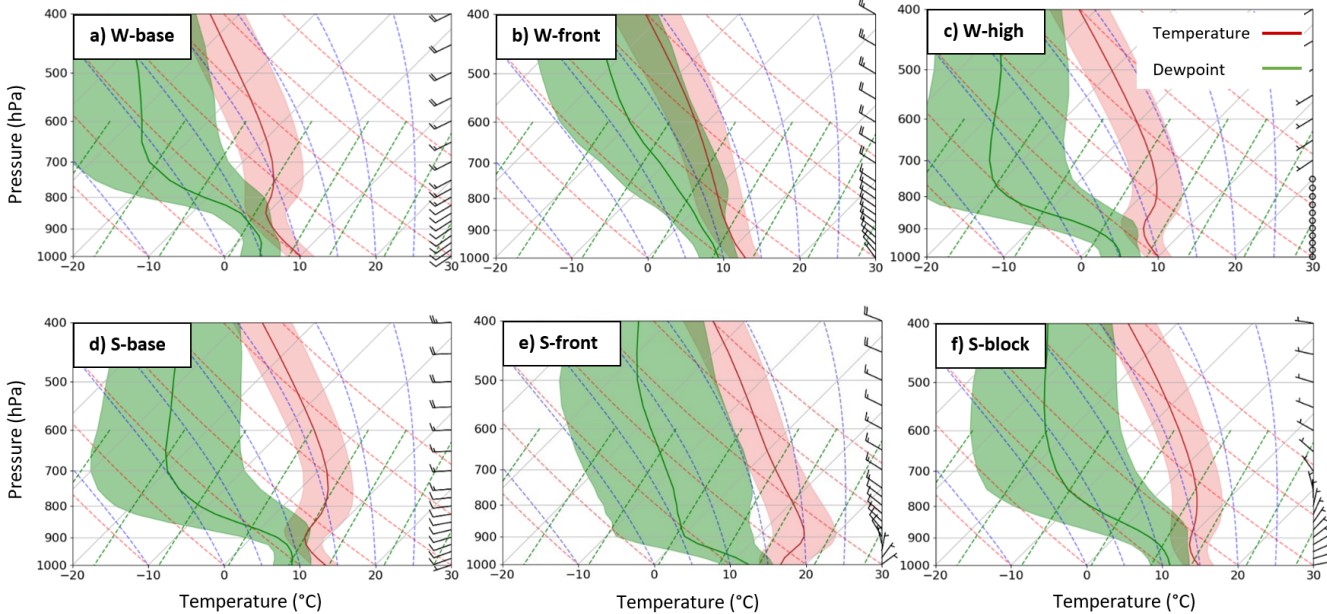

**Figure 4.** Composite soundings for the six clusters (2011-2021). Winter clusters are displayed in the top row while the bottom row shows the summer clusters.

In summary a simple K-means clustering has led to the identification of 6 distinct clusters that exhibit specific synoptic and seasonal meteorological characteristics over CGO. The two baseline clusters are most common, combined they occur ~40% of the time. The two frontal clusters occur ~27% of the time combined. Finally, W-high and S-block occur ~33% of the time, combined (more details can also be found in Table 1).

The 72 hours back trajectories at boundary layer elevation (500 m) (Figure 5) reveal the origin of the air mass being observed at CGO for each of the 6 clusters, largely confirming the synoptic classification. The air mass of both baseline clusters (Figure 5a and d) predominantly originates over the SO, largely free of any terrestrial influence. These also display the greatest displacement compared to the other four clusters (W-base, has the greatest average length of ~3743 km), reflecting the strong westerly winds across the SO storm track. Back trajectories for W-base have a more southerly heading at CGO and are more

likely to have originated at higher latitudes, even occasionally reaching Antarctica; 22% of these trajectories cross the 60°S latitude. The S-base back trajectories have a more westerly heading at CGO with only 2% originating from higher latitudes (crossing 60°S).





Back trajectories for both frontal clusters (Figure 5b and e) suggest a likely terrestrial influence on air masses reaching CGO. During the winter, when the STR is furthest north, the back trajectories still commonly originate over the SO, but can loop
over the continent before reaching CGO. Finally, the back trajectories for W-high (Figure 5c) and S-block (Figure 5f) reflect the weak wind speeds near CGO due to smallest spreads during these synoptic conditions.

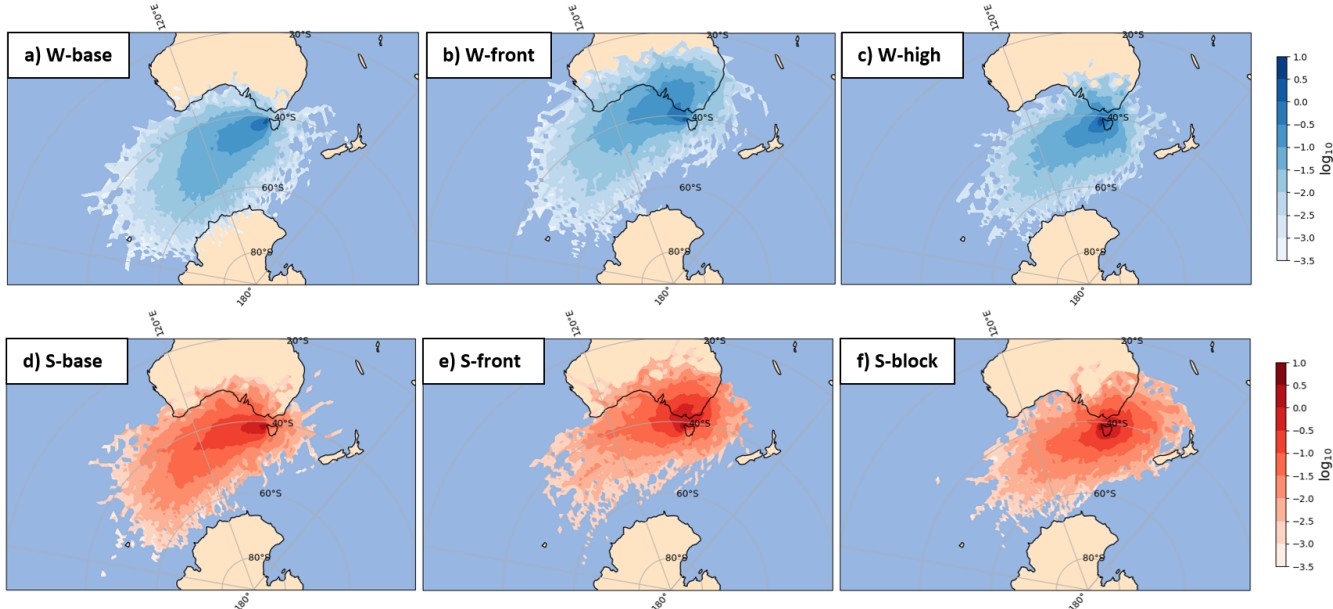

**Figure 5.** Frequency of the distribution of 72hours back trajectories at the altitude of 500 m (boundary layer) for six clusters (2011-2021).

## 4   Air mass characteristics

Having used the ERA5 reanalysis to define the synoptic meteorology at CGO, we now employ these six synoptic clusters to isolate the influence of the meteorology on the CGO records (Table 1). For each of the six clusters, the median CCN and
radon concentrations and mean precipitation intensity and frequency are compiled. We assessed whether these parameters are different between our clusters with the null hypothesis that any differences are only due to random variations. Not surprisingly, W-base is characterised as the most pristine air mass ($N_{CCN}$ 71 cm$^{-3}$, radon 66 mBq m$^{-3}$) having the least exposure to terrestrial influences. S-base, which does pass over Australia more commonly, has around twice the concentration of CCN ($N_{CCN}$ 137 cm$^{-3}$) and higher radon (80 mBq m$^{-3}$). The differences in the median CCN and radon concentration between
these two baseline clusters were found to be statistically significant ($p < 0.05$) using the Whitney U test. Combined, the baseline clusters yield a large seasonal cycle in CCN consistent with previous results (Ayers et al., 1997; Gras and Keywood, 2017; Humphries et al., 2023).





**Table 1.** Median CCN and radon concentration along with their 10th and 90th percentile and the mean precipitation intensity and frequency for the six clusters.

| Clusters (2011-2021) | Number of cases from a total of 8036 | $N_{CCN}$ $(cm^{-3})$ $(10^{th},90^{th})$ | Radon (mBq $m^{-3}$) $(10^{th},90^{th})$ | Precipitation intensity (mm hour$^{-1}$) frequency (%) |
|---|---|---|---|---|
| Winter Baseline (W-base) | 1742 (21.7%) | 71 (28,164) | 66 (35,174) | 0.10 18.4 |
| Summer Baseline (S-base) | 1388 (17.3%) | 137 (47,392) | 80 (33,591) | 0.03 5.8 |
| Winter Frontal (W-front) | 1307 (16.3%) | 223 (69,1061) | 574 (64,3761) | 0.33 30.1 |
| Summer Frontal (S-front) | 925 (11.5%) | 662 (162,2041) | 680 (95,3371) | 0.03 4.8 |
| Winter High pressure (W-high) | 1535 (19.1%) | 126 (36,685) | 197 (55,1361) | 0.02 3.3 |
| Summer Blocking (S-block) | 1139 (14.2%) | 289 (98,949) | 424 (118,1570) | 0.08 6.9 |

Conversely the two frontal clusters are the least pristine, having more than three times greater $N_{CCN}$ than the corresponding baseline clusters. W-front ($N_{CCN}$ 223 $cm^{-3}$, radon 574 mBq $m^{-3}$) is still more pristine than S-front ($N_{CCN}$ 662 $cm^{-3}$, radon 680 mBq $m^{-3}$) for both the CCN and radon concentration. The differences in $N_{CCN}$ between our two frontal clusters were found to be statistically significant ($p < 0.05$) through the Whitney U test, while the difference in radon was not statistically significant ($p = 0.052$). Finally, the two remaining clusters, W-high ($N_{CCN}$ 126 $cm^{-3}$, radon 197 mBq $m^{-3}$) and S-block ($N_{CCN}$ 289 $cm^{-3}$, radon 424 mBq $m^{-3}$), fall in between the extremes. In this case the differences are both statistically significant ($p < 0.05$). Overall, the combined summer clusters have a higher $N_{CCN}$ and radon concentration than the combined winter clusters.



## 4.1 Precipitation

Overall, we find the precipitation rate and frequency for each of the six clusters to be highly consistent with the composite soundings (Figure 4). The differences in mean precipitation between the clusters were found to be statistically significant using the two-tailed Student's t test ($p < 0.05$). W-front has the greatest precipitation intensity and frequency (0.33 mm hr$^{-1}$, 30.1%, respectively) consistent with a weak boundary layer inversion and a high relative humidity up through the free troposphere. Conversely, W-high, which has the smallest precipitation intensity and frequency (0.02 mm hr$^{-1}$, 3.3%), has a strong, shallow boundary layer inversion and a low relative humidity through the free troposphere. S-front, the cluster with the next weakest precipitation intensity and frequency (0.03 mm hr$^{-1}$, 4.8%), has a very strong, low-level inversion.

A strong seasonal difference in the precipitation is also present for the baseline clusters, W-base precipitates (intensity of 0.10 mm hr$^{-1}$ and frequency of 18.4%) at three times the intensity and frequency of S-base (0.03 mm hr$^{-1}$, 5.8%), having a weaker, higher boundary layer inversion and a greater relative humidity through the free troposphere (Figure 4a). We note that the composite W-base sounding is similar to the composite open MCC sounding of Alinejadtabrizi et al. (2024, their figure 2b), while the composite S-base sounding is similar to the composite closed MCC sounding (their figure 2a). Lang et al. (2022) has previously established that open MCC occurs more frequently during the winter in the region up wind of CGO. Overall, higher wintertime precipitation rates are also consistent with the migration of the STR to lower latitudes during the wintertime (Figure 2 top). Manton et al. (2020) reported a negative correlation between precipitation and MSLP over the SO.

Focusing on the baseline air masses, we further explore the inverse relationship between precipitation and N$_{\text{CCN}}$. The higher precipitation rate and lower N$_{\text{CCN}}$ of W-base is consistent with what proposed by Kang et al. (2022); Sanchez et al. (2021); Alinejadtabrizi et al. (2024) regarding the role of coalescence scavenging and wet deposition in cleansing the atmosphere and reducing N$_{\text{CCN}}$. The apparent negative correlation of precipitation and N$_{\text{CCN}}$ is also evident within the two frontal clusters. While W-front and S-front have similar concentrations of radon (574 and 680 mBq m$^{-3}$, respectively), the S-front N$_{\text{CCN}}$ (662 cm$^{-3}$) is more than three times as great as W-front N$_{\text{CCN}}$ (223 cm$^{-3}$) with the W-front precipitation (intensity of 0.33 mm hr$^{-1}$ and frequency of 30.1%) being an order of magnitude greater than that of S-front (intensity of 0.03 mm hr$^{-1}$ and frequency of 4.8%). In the case of the last two clusters (W-high and S-block), however, we observe higher precipitation in summertime (S-block) coinciding with higher N$_{\text{CCN}}$ level. Comparing the back trajectory plots for these clusters (Figure 5c and f), we observe that W-high air masses spend less time over land than those of S-block, presumably acquiring relatively fewer aerosols, on average.

Based on the relationships established in the hourly records of the cloud morphology, N$_{\text{CCN}}$ and the precipitation rate, and the seasonality of the cloud morphology (mesoscale cellular convection) upwind of CGO (Lang et al., 2022), Alinejadtabrizi et al. (2024) hypothesized that a seasonal cycle exists in the baseline precipitation rate at CGO and that wet deposition from this precipitation contributes to the seasonal cycle in N$_{\text{CCN}}$. Combining the two baseline clusters together allows us to examine the seasonal cycle in the baseline precipitation rate at CGO. Figure 6 illustrates the seasonal variation in median N$_{\text{CCN}}$ for our two baseline clusters on the left axis, while the natural logarithm of precipitation and its frequency are shown on the right axes after





filtering out precipitation outliers above the 99th percentile. A clear negative relationship between precipitation and $N_{CCN}$ is
evident, with a correlation coefficient of -0.89 ($p = 0.0001$). This analysis confirms the first part of the hypothesis regarding
the seasonality in the baseline precipitation rate where the mean intensity and frequency of precipitation are 0.02 mm hr$^{-1}$ and
5.6% respectively for summertime (DJF) and 0.09 mm hr$^{-1}$ and 19.8% for wintertime (JJA). This strong negative relationship
between the baseline precipitation and $N_{CCN}$ offers further support to the second part of the hypothesis, but does not provide
conclusive proof. Further, even if the role of wet deposition by shallow convection is established, the importance of this sink
term in the full $N_{CCN}$ budget remains to be quantified relative to the other terms including biogenic production.

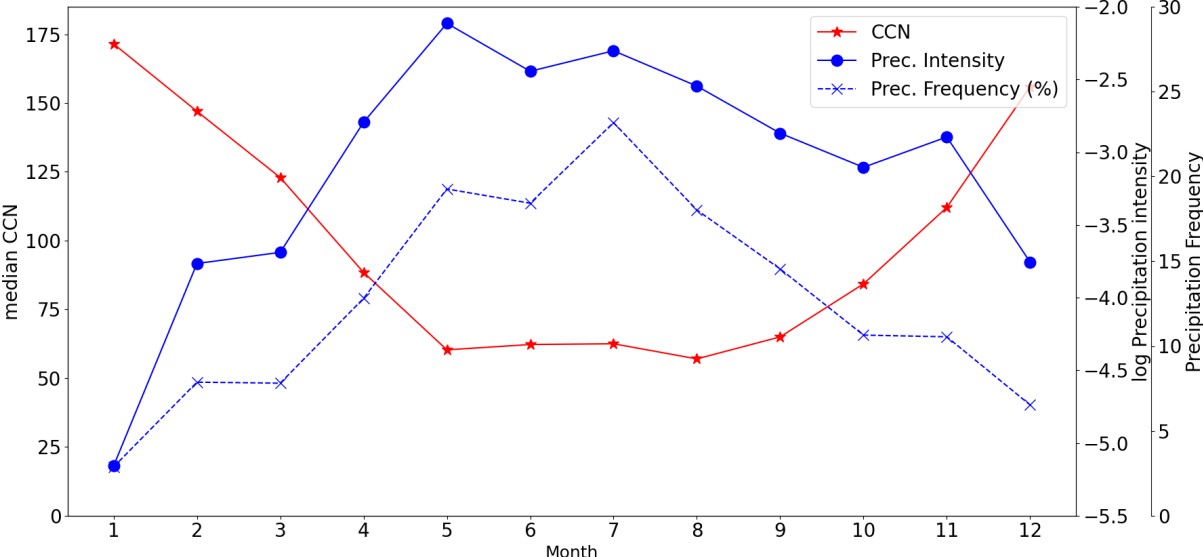

**Figure 6.** Seasonal cycle of median $N_{CCN}$ along with the precipitation intensity and frequency in the two baseline clusters.

## 4.2   Free troposphere entrainment

While potential explanations for W-base being more pristine than S-base are simply there is more biological production in
summer and also there is more precipitation during winter, another explanation could arise from differences in the entrainment
of $N_{CCN}$ from the free troposphere. Is the free troposphere air a greater source of aerosols during the summer? Given that the
radon concentration in S-base is greater than that in W-base (Table 1) and radon is insensitive to precipitation, terrestrial air
masses are presumably having a greater impact on $N_{CCN}$ during the summer baseline cluster, either directly through surface
emissions or through free troposphere entrainment. Kang et al. (2022) found that overall entrainment of free troposphere plays
a larger role than surface sources in controlling cloud droplet number concentration during SOCRATES which was held during
the Austral summer.

To further investigate the role of the free troposphere in the seasonal cycle of $N_{CCN}$, back trajectories were run at the free
troposphere level (2500 m) for both baseline clusters. The difference in the origin of the air masses between W-base and S-



base, especially the frequency of W-base back trajectories subtracted from the S-base (Figure 7), reveals that free troposphere air parcels for the W-base cluster were more likely to have originated over the high latitudes of the SO, while those for the S-base cluster more commonly trace back to lower latitudes near or over land. This is similar to the seasonal behaviour of the baseline boundary layer air masses with the location of the STR helping define the origin of the air mass. While this observation supports the hypothesis that terrestrial influences transported from Australia to CGO through free troposphere entrainment contributes to the observed seasonality in $N_{CCN}$, it is ambiguous as we cannot isolate the behaviour of the free troposphere from the direct surface emissions into the boundary layer air mass.

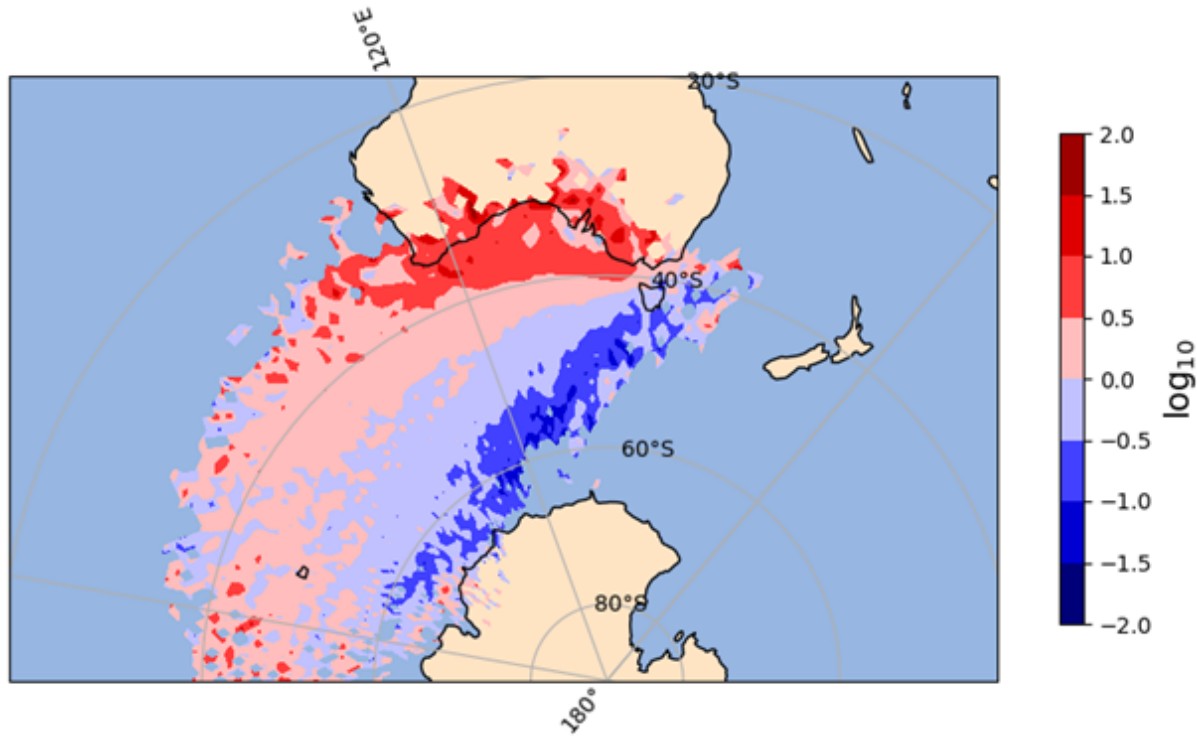

**Figure 7.** Discrepancy in 72 hours back trajectories between the S-base (red) and W-base (blue) clusters at the 2500m elevation (free troposphere).

In an effort to eliminate the potential of direct surface emissions, baseline air parcels were filtered according to their proximity to Australia. If any point of an air parcel's 72 hour back trajectory passed north of 40°S, i.e., gets close to mainland Australia, the air parcel was removed. This filter removed 24% of all W-base hourly records and 55% for S-base. After removing these air masses that pass close to Australia, there remains a statistically significant difference in $N_{CCN}$ between the 'high-latitude' S-base (122 cm$^{-3}$) and the 'high-latitude' W-base clusters (71 cm$^{-3}$). However, no significant difference is observed in their radon concentration (~63 mBq m$^{-3}$). This suggests that air masses originating from high latitudes are not strongly affected by the entrainment of Australian aerosol sources through the free troposphere, regardless of season. The observed difference in $N_{CCN}$ on the other hand could be attributed to variations in sources, such as biogenic production and the sinks e.g., precipitation.




## 4.3 Free troposphere transport across the SO

It is also of interest to further explore the role of free troposphere transport across the wider SO within the framework of the synoptic clusters. An analysis of the 48-hour forward trajectories at the free troposphere elevation (2500 m) for all six clusters has been presented in figure 8. Here after, our analysis is limited to the W-front and S-front clusters, as frontal air masses have been observed to transport dust and bushfire plumes across the high latitudes of the SO (e.g., Attiya and Jones, 2022; Fox-Hughes, 2023). These air masses also have spent the greatest amount of time over Australia, thus having high radon concentrations (Chambers et al., 2017).

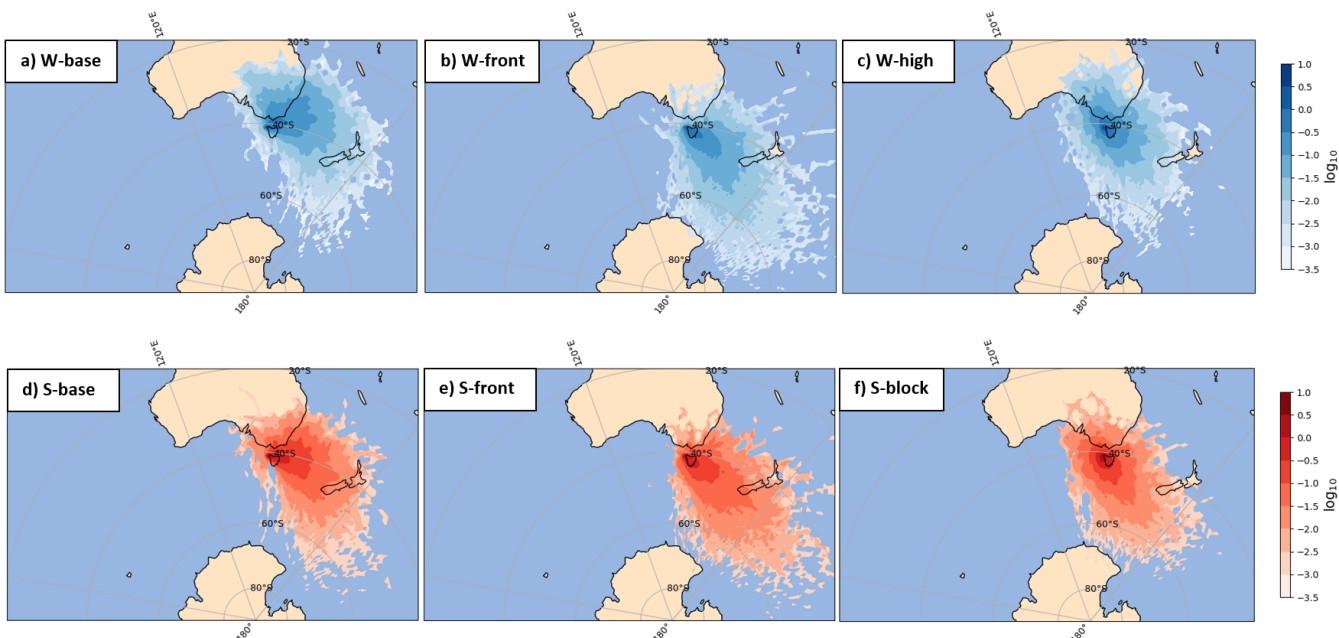

**Figure 8.** Frequency of the distribution of 48 hours forward trajectories at 2500 m (free troposphere) level for six clusters (2011-2021).

The analysis of the forward trajectories reveals a notable distinction: the W-front cluster extends furthest poleward, with more air parcels (~43%) reaching high latitudes (55°S) compared to the S-front cluster (~28%). Given the high radon concentration of these frontal air masses, it follows that they contribute to higher radon concentrations over the higher latitudes of the remote SO, especially during the winter season. This aligns with the findings of Chambers et al. (2018), who observed a seasonal cycle in the radon concentration over Macquarie Island (54.5°S, 159.0°E) characterized by a winter maximum (100 mBq m$^{-3}$) and summer minimum (75 mBq m$^{-3}$) based on the 4 years of observation (2013-2017). This seasonal difference in the extent of the forward trajectories again suggests that the location of the subtropical ridge moderates the transport of free troposphere air across the SO. During the summer, high-pressure systems can commonly set up poleward of Tasmania, acting as a barrier to the long-range transport of continental air masses across the free troposphere. In the winter, the extended frontal systems have the potential to draw the continental air masses more freely across the SO.



## 5   Relationship to CGO baseline air masses

Based strictly on the clusters produced from the ERA5 thermodynamics through the lower free troposphere, we have defined the W-base and S-base clusters. These clusters are independent of the established CGO definition(s) of baseline conditions in the literature. We now explore the consistency between these two different methods of defining baseline (Figure 9). This analysis helps to demonstrate the robustness of our findings with the CGO baseline definitions. Moreover, it highlights the potential significance of synoptic meteorology in understanding baseline conditions.

The primary criterion for CGO baseline is that the local surface wind heading must be between 190°and 280°, i.e., a south-westerly to westerly heading. Over our 11-year record (twice per day), we have 3478 hourly records that meet this criterion using ERA5 winds. We acknowledge that there may be discrepancies between the measured local winds and the ERA5 winds. Of these 3478 hourly records, ~75% are classified as W-base and S-base. Of the remaining 25%, roughly half (12%) come from the W-high cluster. This single criterion for defining baseline is known to be weak with median $N_{CCN}$ at 90 cm$^{-3}$ and
radon at 71 mBq m$^{-3}$ (Figure 9a), indicating that continental air masses are being sampled. Looking at this conversely, ~91% of the W-base hourly records and ~76% of the S-base hourly records meet this primary CGO baseline definition. The clustering of ERA5 records is highly consistent with this CGO criterion.

To reduce the influence of terrestrial air masses, it is common for the CGO baseline criteria to be further constrained, removing air masses with high radon concentrations. This radon threshold has become more and more strict over time, reflecting an
increasing appreciation of the potential influence of free troposphere entrainment. We have chosen to employ one of the earlier (weaker) radon thresholds of 150 mBq m$^{-3}$, which still proves to be highly effective, reducing the value of median $N_{CCN}$ from 90 cm$^{-3}$ to 85 cm$^{-3}$ and radon concentration from 71 mBq m$^{-3}$ to 61 mBq m$^{-3}$. This additional constraint removes nearly 20% of the records. We find that nearly 81% of the remaining 2728 hourly records would now be classified as our baseline clusters (W-base and S-base). The W-high cluster still accounts for over 11% of these records (Figure 9b). Most of the records
filtered out by this second threshold came from S-base again highlighting the increased potential for both free troposphere entrainment and direct surface emissions of radon during summer (Figure 9c). It should be noted that a stricter radon threshold (80 mBq m$^{-3}$) has been examined, and the results (not shown here) indicate that it does not affect the median $N_{CCN}$. However, the median radon has decreased from 61 to 51 mBq m$^{-3}$. Also, the percentages of contribution of each cluster do not change significantly.

A third criterion based on wind speed can be further applied to the CGO baseline definition. Following the literature, we set a minimum wind speed of 5 m s$^{-1}$ (Jimi et al., 2007), which removes a further 7% of the record. W-base and S-base comprise nearly 85% of these remaining records. Both CCN and radon concentration do not show changes by this last constraint (Figure 9d). Not surprisingly, W-high records, which have low surface wind speeds, are primarily removed by this last constraint (Figure 9e). Nevertheless, over 9% of this highly constrained CGO baseline record still come from W-high rather than baseline
clusters.




It is interesting to directly compare efficiency of the two different methods of defining baseline conditions. The most constrained CGO definition, using all three thresholds, produces a median value of 84 cm$^{-3}$ for $N_{CCN}$ and 61 mBq m$^{-3}$ for radon. This method makes no distinction for winter or summer. Conversely the original W-base cluster produced values of 71 cm$^{-3}$ and 66 mBq m$^{-3}$ for $N_{CCN}$ and radon, respectively, while the S-base cluster produced values of 137 cm$^{-3}$ and 80 mBq m$^{-3}$ (Table 1). This again suggests that the seasonal changes in the meteorology are having a direct effect on the seasonal cycle of $N_{CCN}$ as observed at CGO.

**Figure 9.** Comparison of derived clusters with CGO baseline criteria using three criteria: (a) wind direction (190°-280°), (b) wind direction plus radon <150 mBq m$^{-3}$, (c) cases removed when adding radon criterion to (a), (d) wind direction and radon plus wind speed >5 m s$^{-1}$ and (e) cases removed when adding wind speed criterion to (b).



## 6 Discussion and Conclusion

Our study provides new insight into the impact that the synoptic meteorology has on the observed seasonality in $N_{CCN}$ at CGO. Specifically, we explore how the seasonality of the synoptic meteorology affects precipitation, which acts as a sink through wet
deposition, and the free troposphere transport of terrestrial air masses, which acts as a source of $N_{CCN}$ through entrainment.

Utilizing clustering analysis on ERA5 thermodynamic data, we observed a strong seasonal cycle in the synoptic meteorology. Specifically, three synoptic clusters (W-base, W-front and W-high) were found to be more prevalent during the winter months (JJA) while another three (S-base, S-front and S-block) were more common in summertime (DJF). The baseline clusters, W-base and S-base, are characterized by south westerly winds at the surface, with a deeper boundary layer inversion in winter
suggesting more frequent precipitation from shallow MABL clouds. The frontal clusters, W-front and S-front, feature strong north westerly winds through the free troposphere, with W-front exhibiting higher relative humidity. W-high displays near zero wind speed, on average, and minimal precipitation, while S-block is characterized by low wind speed, anti-cyclonic atmospheric conditions. Not surprisingly, the W-base cluster is characterized as the most pristine air mass, while S-base, which occasionally passes over Australia, exhibits around twice the concentration of CCN and 20% higher radon concentration. These
findings highlight a large seasonal cycle in $N_{CCN}$ consistent with previous research (Ayers et al., 1997; Gras and Keywood, 2017; Humphries et al., 2023). Conversely, the two frontal clusters are identified as the least pristine, with $N_{CCN}$ more than three times greater than the corresponding baseline clusters. The two remaining clusters, W-high and S-block, fall between the extremes. Overall, the combined summer clusters exhibit higher CCN and radon concentrations than the combined winter clusters.

Our analysis reveals an inverse relationship between precipitation and $N_{CCN}$ during both the baseline and frontal clusters, highlighting the role of coalescence scavenging and wet deposition in cleansing the atmosphere and reducing $N_{CCN}$ over the SO (Kang et al., 2022; Sanchez et al., 2021; Alinejadtabrizi et al., 2024).

Our analysis of the role of free troposphere entrainment at CGO was inconclusive. While the back trajectory analysis reveals that S-base is more commonly affected by terrestrial (Australia) influences, it was not possible to isolate free troposphere
entrainment of $N_{CCN}$ from direct surface emissions. Either way, however, the S-base air mass is more frequently affected by terrestrial sources than the W-base air masses, again revealing that other sources are contributing to the seasonal cycle in $N_{CCN}$ as observed at CGO other than biogenic production. Looking more widely at free troposphere transport across the SO, forward trajectories of the frontal clusters were also examined. The forward trajectories of the W-front cluster were found to extend further poleward than those of the S-front cluster, consistent with the seasonality of radon observations at Macquarie Island
characterized by a winter maximum and summer minimum (Chambers et al., 2018), underscoring the potential contribution of the entrainment of free tropospheric continental air masses into higher latitudes.

On average the characteristics of our two baseline clusters are consistent with those of the traditional CGO baseline air mass. Our analysis finds that the wintertime baseline precipitation is approximately three times greater than that during the summer, helping make the wintertime baseline air mass more 'pristine' through wet deposition. An examination of the transport of



the overlying free troposphere air also finds a distinct seasonal cycle with terrestrial air masses more commonly passing over kennaook/Cape Grim during the summer season, when the subtropical ridge is furthest poleward. The entrainment of this terrestrial free troposphere air into the boundary layer will also contribute to seasonal cycle in $N_{CCN}$.

Our analysis of both forward and back trajectories reveals that overall, during the winter, when the STR resides further north, toward the equator, CGO exhibits heightened connectivity to the higher latitudes and Antarctica. Conversely, during summer

when the STR shifts poleward, this connectivity is weakened. In effect, the STR acts as a barrier. This seasonal modulation underscores the significant influence of large-scale meteorological patterns on air mass observed at CGO.

With respect to our understanding of the CGO baseline air mass, two salient points arise. First, the current criteria for defining the CGO baseline air mass includes a non-negligible percentage from the W-high synoptic class. Second, and more importantly, there are significant seasonal differences in the boundary layer structure, precipitation and air mass origin (both boundary layer

and free troposphere). Echoing the conclusions of Quinn and Bates (2011), a full understanding of the $N_{CCN}$ budget over the Southern Ocean is far more complex than simply an understanding of the biogenic production. In particular, it is essential to understand the role of precipitation from shallow convection across the SO (Siems et al., 2022; Alinejadtabrizi et al., 2024).

*Data availability.* The $N_{CCN}$ measurement, analyzed during the current study are available in the World Data Centre for Aerosols [http://www.gaw-wdca.org/]. The ECMWF-ERA5 reanalysis datasets are available through the Copernicus Climate Change Service Climate Data Store

[https://cds.climate.copernicus.eu]. The precipitation data can be obtained by contacting [climatedata@bom.gov.au]. The radon data is available from the World Data Centre for Greenhouse Gases (WDCGG) [https://gaw.kishou.go.jp/] and from Alastair Williams from Australian Nuclear Science and Technology Organisation (ANSTO).

*Author contributions.* Alinejadtabrizi performed the data analysis and prepared the original draft of the paper. All co-authors provided editorial feedback on the paper. All co-authors read and approved the final manuscript.

*Competing interests.* At least one of the (co-)authors is a member of the editorial board of Atmospheric Chemistry and Physics. Authors have no other competing interests to declare

*Acknowledgements.* This research has been supported by Securing Antarctica's Environmental Future (SAEF), a Special Research Initiative of the Australian Research Council (SRI20010005) and also by the Australian Research Council (ARC) Centre of Excellence for Climate Extremes (CE170100023) and the ARC Discovery Projects (DP190101362). Continued support for the kennaook Cape Grim Program

from the Australian Bureau of Meteorology and Commonwealth Scientific and Industrial Research Organisation (CSIRO) is also gratefully acknowledged.



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
