# Peer review of "Contributions of the synoptic meteorology to the seasonal CCN cycle over the Southern Ocean"

_EGUsphere, 2024_

## Author Response (AR1)

School of Earth, Atmosphere and Environment,

Monash University, VIC 3800, Australia

Prof. Greg McFarquhar,

Atmospheric Chemistry and Physics

October, 2024

Re:

We genuinely appreciate this opportunity to improve our manuscript in response to the constructive reviews (and encouragement) from the two reviewers. We thank the reviewers for their time, effort, and ideas.

In response, we have carried out major revisions as summarised below.

- We have reordered and combined the figures and related discussions to improve readability.

- We quantified the environmental context for each cluster (specifically the estimated inversion strength and the total column water vapour or precipitable water and the inversion height) to improve our discussion.

- We have further clarified our inconclusive discussion on the free troposphere entrainment and its role in the observed seasonality in CCN over the CGO.

- We have analysed the height distribution of the back trajectories to show the large-scale subsidence over our study area, however we acknowledge that ERA5 can't capture the entrainment as it is a sub-grid scale process.

- We removed the forward trajectories analysis and discussing Macquarie Island data, focusing more on the main scientific questions over the CGO.

A detailed point-by-point response follows.

**Reviewer 1.**

*This paper generally classified 6 synoptic conditions using K-means and looked into each corresponding sounding and pressure field. Local current precipitation is confirmed to be anti-correlated to the population of CCN. The authors have also done a back-and-forward trajectory analysis to check the history and future of tropospheric transportation. This method is also compared to the previous criteria to check each corresponding influence on the clusters.*
*The most significant conclusion may be that the connections between the focused SO region and the polar area are strong during the winter when the STR is close to the equator, but weakened for the poleward shift of STR during the summer. Implicitly, this summertime poleward transition constrained the precipitation, consistent with the higher CCN population at CGO. Furthermore, the wintertime southern wind is consistent all through the FT so that in general the air mass originates from the south more, even close to the Antarctic. The most unique part of this paper is about using Radon as a reference factor to isolate the influences from the FT transportation and precipitation. Even though the eventual results of this section are inconclusive, the statistic itself is worthwhile to be published and discussed for adding novelties to this paper. However, there are some pivotal gaps in the logic line thus Major revision is suggested to help make the paper flow better.*

> Thank you for your thorough review of our manuscript. We appreciate your understanding of the complexities involved in our study. As you correctly noted, some of our results, particularly those related to the role of free troposphere entrainment, remain inconclusive. Despite these uncertainties, we have endeavored to extract as much insight as possible from the available datasets, aiming to illuminate the potential impact of synoptic meteorology—through both precipitation and free troposphere entrainment—on the CCN population at the Cape Grim Observatory (CGO).

While some conclusions are still open to interpretation, we believe that presenting these findings contributes valuable perspectives to the ongoing discussions in this field. We also appreciate your feedback regarding the logical flow of the manuscript. We have committed to revising the manuscript to enhance clarity and ensure that our findings and their implications are communicated more effectively. Your suggestions have been carefully considered as we worked to improve the structure and coherence of the paper, aiming for a clearer and more straightforward presentation of our results and conclusions.

**Major comment:**

*Recommend labelling Figure 3 to be Figure 1. The discussion of Figure 1 is primarily based on Figure 3, thus shifting order can help the readers a lot if specific schemes are shown first.*

In response to your recommendation, we have changed the order of the figures to enhance the readability and understanding of our manuscript.

*Section 2 largely lacks a detailed description of precipitation observation. Lines 103-104 are too limited for the interpretation/evaluation of Figure 6. What are the observed precipitation frequency and intensity time resolution; what is precipitation frequency defined; what are the data quality controls and raw data processing processes (acknowledge the mention of the 90th outlier removal in a later section); how large the uncertainties are; what is the precipitation phase; why using the MEAN precipitation to relate to the MEDIAN of CCN; can Figure 6 has variations/ranges shown, e.g. in shading, together with the mean/median value of each month.. Briefly mentioning some of the info above would help validate Figure 6 given using rain gauge data itself can be an advantage compared to the reanalysis data.*

We have addressed your concerns by providing additional details on the precipitation data (e.g., lines 113 to 119) in the revised manuscript. Specifically, we clarified the frequency of precipitation and explained why we use the mean precipitation, in contrast to the use of the median for CCN. Since 88% of the time (hours) there is no precipitation, the median value is 0, which makes the mean more appropriate for capturing the range of precipitation events. In contrast, the median was used for CCN data to mitigate the influence of outliers.

Additionally, we have updated Figure 6 (now Figure 5) to incorporate all cases within our two baseline clusters. We now include the 25th and 75th percentiles for CCN data, providing a clearer depiction of the distribution. For precipitation, we added the 95th percentile, though the 5th percentile remains at 0, which would yield an infinite log value.

*Section 3 serves as the most fundamental part of this paper, which can go further into depth. Recommend adding wind field and front/ridge location on Figure 3 to, 1) help the description of Figure 4 which refers to the consistent surface wind direction and speed at CGO, 2) help the reader see the relative locations of the surface front system and the CGO, which is important for convergence/divergence flow and can be used to explain the Radon concentration as well as the spread extent in history trajectories (related to the wind speed according to Line 181). Recommend the description of inversion from altitude, depth, and strength perspectives, and a more quantitative description can be added in for lines 150-165. For example, it can start from one cluster, mentioning the specific numbers of inversion layer location and depth, and maybe also the estimated inversion strength, boundary layer depth, potential cloud layers locations, where is the referred FT, and how large the Relative Humidity is through the FT. Numbers are vital for intra-comparison in Figure 4 between clusters. Also, in Line 161, "the relative humidity is greatest for W-front suggesting heavier precipitation rate", the small gap between the two temperature profiles illustrates a more humid condition but does not necessarily suggest a heavier precipitation rate. The mention of open and closed MCC by cross-citing is wonderful and can be shifted here from Section 4.1 which is about precipitation.*

We have added wind vectors (10m wind) to the figure (New Figure 1) to provide clearer visualization when comparing the composite soundings with the composite MSLPs.

We explored using the composite field to define a front based on the gradient of potential temperature or equivalent potential temperature of the 850 hPa (Thomas and Schultz, 2019) but believe the results can easily be misinterpreted. As the clustering algorithm is centered on CGO, the physical coherence of the meteorology decays with distance from CGO. In particular, the signal for the front (at any threshold) decays rapidly at higher latitudes over the storm track, where there is great variability in the location of the cyclone. Similarly constructing a composite of frontal locations becomes dispersed at high latitudes. Even near CGO the frontal position – and orientation (east/west vs north south) – experiences a high degree of variability. The summer baseline composite – when the subtropical ridge is at these higher latitudes - is particularly variable.

We can put the trough (and ridge) axes onto the MSLP, but this is different than the frontal position.

[Figure]

Potential temperature gradient at 850 hPa (°K/100km) for 6 clusters

[Figure]

Equivalent potential temperature gradient at 850 hPa (°K/100km) for 6 clusters

Here, we present an example of six cases within the S-base cluster, illustrating the significant variability in frontal locations within the same cluster. This diversity highlights why creating a composite of these frontal positions is not feasible.

[Figure]

[Figure]

We have revised lines 163-184 to include the specific measurements of inversion height and estimated inversion strength (EIS). Furthermore, as suggested, we have relocated the discussion comparing the baseline clusters with the open and closed MCCs to this section (lines 170-176) to better align with the synoptic meteorological differences within our clusters. Moreover, we have removed the mention of relative humidity from this section, as you correctly pointed out that higher relative humidity does not necessarily correlate with higher precipitation. Instead, we have incorporated the discussion of relative humidity into Section 4.1, where we discuss the precipitation in greater detail. In this section, we have quantified relative humidity and total precipitable water to better support our discussion (e.g., lines 226-236)

Thomas, C. M., and D. M. Schultz, 2019: What are the Best Thermodynamic Quantity and Function to Define a Front in Gridded Model Output?. Bull. Amer. Meteor. Soc., 100, 873–895, https://doi.org/10.1175/BAMS-D-18-0137.1.

*Section 4: reading until now, I recommend combining Figures 1 and 2 in one figure to show the consistency between each other, and adding one new figure showing the correlation plot between Nccn and Radon concentration and that between Nccn and Precipitation, this can be a key figure for the follow-up discussions in Section 4-6. Looks like the correlation has been done already given some of the P-values are provided, plots may better show how related they are.*

As per your suggestion, we have combined Figures 1 and 2 into a single figure to better illustrate the consistency and relationship between the seasonal migration of the subtropical ridge (STR) and the clustering results (now called Figure 2).

Regarding the additional correlation plots between $N_{CCN}$ and Radon concentration and between $N_{CCN}$ and Precipitation, we have generated these plots: the correlation coefficient between Precipitation and CCN is very low (0.007) and not statistically significant (p = 0.9). The scatter plot does not show any clear trend, likely due to highly intermittent nature of precipitation over the region as discussed in Alinejadtabrizi et al., (2024). We also looked at the lagged correlations of the precipitation and CCN, but it remains statistically insignificant.

[Figure]

Scatter plot illustrating the correlation between CCN and precipitation in base line clusters.

While the correlation between Radon and CCN is statistically significant (correlation coefficient = 0.4, $p \approx 0.0001$), the scatter plot does not provide a clear visual representation that would effectively contribute to the discussion in the main body of the manuscript. The significant correlation suggests an association, but due to the scatter in the data, the relationship is not strong enough to warrant inclusion as a primary figure.

[Figure]

Scatter plot illustrating the correlation between CCN and radon in base line clusters (for the CCN and radon concentration less than 1000)

We have discussed these relationships in more detail within the manuscript (e.g., line 272) to provide further clarity. Given the lack of strong visual clarity in the plots, we have opted not to include them in the main body.

*How is Radon distributed through the troposphere, do the air mass coming from the boundary layer and FT above the Australian continent have the same Radon concentration? If not, the altitude of air mass back history would be of importance here, before any discussions about staying over terrestrial in history for long is related to the higher Radon concentration.*

We are unaware of any observations of the vertical distribution of radon through the atmosphere, either over the Southern Ocean or over Australia, that would be suitable for this climatological study.

*The largest concern though comes from the interpretation of Figure 7 and the descriptions in Section 4.2. This needs further clarification. The log10 refers to the log10 scale of the "W-base trajectories subtracted from the S-base"? Does this mean the subtraction results have to be positive to be able to be "logged"? How would the back trajectory starting from 2500m above the GCO be directly useful*

*for surface CCN from the understanding of FT entrainment? I assume the logic is that, if the FT air masses originate more from above the Australian continent, then the FT may contain more aerosols and thus can be a strong source for the surface CCN budget. However, are there any conditions that have to be met so that the 2500m air subsides into the boundary layer? What are the roles of the two below scenarios respectively, 1) an air mass originates from above the Antarctic at, for example, 2500m, and gets transferred into the boundary layer of GCO, and 2) as Figure 7 shows, some air masses from above the Antarctic during the winter travel and arrive at the 2500m right above the GCO. In particular, the 2500m starting level shows more FT continental aerosol information but how can this be used for the surface/below clouds CCN budget discussion without details of discussion of "processes right above the GCO between the surface and 2500m" such as mixing/exchanging and cloud processing? For Line 249, in Kang et al. (2022), FT CCN was quantified using the UHSAS measurement in the FT, and for surface sources, only the wind-oriented primary CCN is quantified in the budget. However, the logic in this paragraph originates from (line 242) whether the less pristine are caused by/related to the biological production in the summer. Could Kang's (2022) paper be used here to support/debate the conclusion? Instead, Kang's conclusion about FT entrainment influences the NCCN more can be heavily related to the surface biological production of, e.g., DMS.*

We fully respect and understand the need for further clarification in interpreting Figure 7 and the associated discussion in this section. We have addressed your comment by splitting it into smaller parts, and by doing so, we hope to enhance the clarity of this section.

The log10 transformation is applied to the individual normalized trajectory counts (frequencies) for both S-base and W-base clusters before performing the subtraction. Therefore, the operation being represented is the difference between the logarithms of the trajectory frequencies (log10(S-base) - log10(W-base)). Mathematically, this is the same as the log of the ratio of the frequencies. We have adjusted the colour bar caption in this to clarify this process and to avoid any further confusion. Additionally, following suggestions from another reviewer, we have included a similar plot for the boundary layer back trajectories, which show consistent patterns—air parcels originate more frequently from lower latitudes in summer, while in winter, they primarily originate from higher latitudes.

The original Figure 7 cannot be used to distinguish between these two scenarios that you raised, as it only pertains to parcels at 2500 m over CGO at the start of the back trajectories, not simultaneous parcels in the free troposphere and the boundary layer. We accept that we have no knowledge of either free troposphere entrainment or potential cloud processing that could affect any relationship between free troposphere and surface CCN concentrations. You are correct that we are simply exploring the potential role of free troposphere transport within the framework of this synoptic typing.

As you correctly pointed out, our assumption is that entrainment from FT occurs, and now the question is about the sources of air masses in the FT level. If these air masses originate more frequently from above the Australian continent, they likely contain higher aerosol concentrations, which could significantly contribute to the surface CCN budget. However, determining the exact conditions under which the 2500 m air entrain into the boundary layer required the cloud processing and entrainment information which is not available over the CGO. We further, examined plots showing the distribution of frequency of the 72-hour back trajectories' heights (added as supplementary materials-Figure A1) which demonstrate the subsidence of air parcels, however, ERA5 do not explicitly capture entrainment processes as they are sub-grid scale phenomena.

Therefore, our analysis relies on the assumption that FT entrainment acts as a potential source contributing to the observed seasonality of CCN based on the available datasets. We added lines 287 to 295 in an effort to make this discussion clearer.

In response to the comment regarding the Kang's paper, we have also clarified our discussion of Kang et al. (2022) in the revised manuscript. We acknowledge that Kang et al. (2022) emphasize the importance of free troposphere entrainment from the biogenic productions,

however, we focussed on the entrainment from continental sources. We have revised our paragraph to reflect this nuance (e.g., lines 275 to 278)

*In general, Figure 5 is about the advection history. While Figure 7 is also an advection history (FT though), it is analyzed as a local source for the surface CCN budget at GCO. Then the gap question would be, how efficient are the advections and the local FT entrainment? Without the filling of this gap, Figure 7 only talks about "the potentials" of FT air mass feeding the surface CCN.*

As we mentioned in a previous comment, we do not have direct data on entrainment or advection to assess the efficiency of advection and local FT entrainment over CGO. Nor can we get this from the reanalysis. Our analysis was conducted based on the assumption that entrainment is occurring, with the focus on understanding the differences in the origin of the air masses. As you correctly pointed out, this approach does not allow us to confirm the role of FT entrainment in feeding the surface CCN. Therefore, all we can discuss is the potential contribution of FT air masses to surface CCN. We tried to clarify this in lines 287 to 295.

*Section 4.3 looks quite independent from the other part of the paper, in particular since the Macquarie Island data are not shown/heavily discussed. Suggest a removal of this section/shift to supporting materials so that the most important scientific question (precipitation/FT transportation) can be focused on using the 11 year data from the GCO.*

We agree with your reservation about this section and have removed this section from the revised manuscript.

*Maybe briefly mention the reasons for some methods that are used. Why is 72h chosen for back trajectories instead of 35h or 120h? Why is 2500m chosen as a reference for FT? Why are two-tail-student-t-tests used for precipitation while Whitney-U tests used for CCN and radon?*

We selected 72 hours for the back trajectories to demonstrate the connectivity between our study area and both lower latitudes (the continent) and higher latitudes (Antarctica). Additionally, 72 hours aligns with the typical time scale between cyclones in the Southern Hemisphere, where approximately two cyclones pass through each week. We added a note in this regard in lines 190 to 193.

We selected 2500 m as the free troposphere level based on field observations from Lang et al. (2021), which reported that boundary layer clouds, specifically open and closed MCCs generally form below 2.5 km. By choosing 2500 m, we aimed to ensure that our analysis captures air parcels above these shallow cloud formations. We have clarified these points over line 280.

We used the two-tailed Student's t-test for precipitation because we analyzed the mean precipitation, and the t-test is appropriate for comparing means between two groups. For CCN and radon, where we analyzed the medians, the Mann-Whitney U test was used as it is more suitable for medians. We have added a note in this regard in lines 226 and 212 respectively.

**Minor comments:**

*• Suggest making the abstract only one paragraph, with more quantitative descriptions. For example, what is the referred "deeper boundary layer" (line 9); specifically how do STR moderate the advection of air masses (line 15).* We have modified the abstract accordingly.

*• Line 49-50, "since new particle formation is rare in the MABL" turns out to be a fundamental basic assumption for this study, which will need citations to support. Zheng et al. 2021 actually states that there are observed new particle formation (NPF) in the remote MABL. Zheng et al. 2018 indeed mention that the NPF events within remote MBLs like the ENA are infrequent, but this is only done through citing other papers, which should be cited instead.* Related references have been added.

*• Figure 2 caption, what are the hollow circles on the plot?* They are the outliers. We added this to the figure caption.

*• Line 60 grammar check.* The paragraph has been revised for better clarity and flow.

*• Line 63 grammar check.* The paragraph has been revised for better clarity and flow.

*• Line 40, ".. marine biological sources predominantly govern Nccn during the summer…, ,multiple elements contribute to the CCN throughout the year…" recommend specifically mention what are the multiple elements.* Done (line 38).
*• Similarly Line 43, "various other sinks and sources influence the CCN budget" what the "various other".* Done (line 41).
*• Line 43, "coarse mode sea salt", coarse mode normally refers to aerosols larger than 1um, delete the "coarse-mode", sea salt plays a crucial role in CCN…* Done (line 38).
*• Line 43-44, recommend adding in related publications about SO aerosols, for summer: Fossum et al. 2018 Scientific Reports, about seasonal variations of CCN. Humphries et al. 2023 ACP, Niu et al. 2024 JGR-A, etc.* Done (line 42).

**Reviewer 2.**

*This study identified six main synoptic clusters affecting the Cape Grim Observatory (CGO) and examined the impacts of synoptic meteorology on the observed seasonality of cloud condensation nuclei (CCN). The seasonal cycle of these clusters closely relates to the subtropical ridge's migration, resulting in different wind patterns and precipitation for each cluster. Backward trajectory analysis shows that two baseline clusters predominantly originate from the Southern Ocean, with less terrestrial influence, a finding further supported by radon measurements. For these two baseline clusters, CCN concentrations are inversely related to precipitation intensity and frequency. The study also examined the role of free tropospheric transport using back-and-forward trajectory analysis.*

*Overall, this paper presents an interesting narrative on the seasonality of CCN from the perspective of synoptic meteorology and enhances our understanding of CCN variability over the Southern Ocean. I recommend accepting this paper with the major revisions suggested below.*

> We sincerely appreciate your positive feedback on our study and your recognition of its contribution to understanding CCN variability over the Southern Ocean. We have carefully considered all of your suggestions and have made the necessary revisions to improve the clarity of our manuscript. Your insightful comments are invaluable in refining our analysis, and we believe that the revisions have strengthened the overall narrative of the paper.

**Major comment:**

*Line 227-240, Figure 6: A negative correlation was found between precipitation and CCN. However, the seasonality of CCN also reflects the effect of seasonal variations in sources (e.g., biogenic aerosols are higher in the austral summer and lower in winter). I'm curious if the authors can further isolate the effects of the source and sink by filtering the CCN data based on precipitation rates. For example, what would the CCN in the current Figure 6 look like for non-precipitating and precipitating cases, respectively? For the non-precipitating cases, the seasonality of CCN would mainly be due to sources. On the other hand, the seasonality of precipitating cases would include the effect of both source and sink. I suspect the seasonality of CCN for non-precipitating cases would resemble that in the current Figure 6 (but with higher values). It is also likely that CCN for non-precipitating cases would negatively correlate with precipitation due to coincidental lows and highs. Moreover, the ratio of the CCN values between precipitating and non-precipitating cases might be indicative of the role of precipitation in controlling the seasonality of CCN.*

> We appreciate your suggestion to further isolate the effects of source and sink processes by filtering the CCN data based on precipitation rates. Following your recommendation, we analyzed CCN concentrations for both precipitating and non-precipitating cases. As you correctly anticipated, the seasonality in CCN concentrations during non-precipitating events (red star-line in the attached figure) are slightly higher (median of 161 for summer (DJF) and 62 for winter (JJA)). Conversely, CCN values are lower during precipitating conditions (the green star-line in the attached figure) (median of 134 for DJF and 57 for JJA), which we attribute to aerosol washout from precipitation.

> However, we believe that analyzing CCN based on hourly precipitation data may not provide a fully accurate representation, given the highly intermittent nature of precipitation in our study area as discussed in Alinejadtabrizi et al., (2024). Specifically, open MCCs often produce intermittent precipitation in one hour, followed by non-precipitating periods in the subsequent hours. Following figure from Lang et al., (2021), show a time series of the precipitation based on the CAPRICORN 2016 Field Campaign, which illustrate the nature of the precipitation over our study area. This discontinuity could obscure the impact of precipitation on CCN, as the washout effect may still be present even when the event is categorized as non-precipitating. To avoid potential misinterpretations, we have opted to retain our original approach in this section,

which we believe offers a more consistent analysis of the relationship between CCN and precipitation.

[Figure]

[Figure]

Time series of surface level precipitation, temperature, and relative humidity during a case study within the CAPRICORN 2016 field campaign (Lang et al., 2021 (their figure 2d))

Lang, F., Huang, Y., Protat, A., Truong, S. C. H., Siems, S. T., & Manton, M. J. (2021). Shallow convection and precipitation over the Southern Ocean: A case study during the CAPRICORN 2016 field campaign. Journal of Geophysical Research: Atmospheres, 126(9), e2020JD034088.

*Section 4.2: This subsection focuses on free tropospheric entrainment and the backward trajectories were run at 2500m. If the focus is on the contribution from the free troposphere to the boundary layer (where CGO CCN measurements were made), why weren't the backward trajectories initialized from the boundary layer instead? In addition, the trajectory analysis in this study mostly focuses on the spatial distribution of the trajectories. How do the trajectories vary vertically?*

To address this, we ran additional analyses at the boundary layer level and found that the results are largely consistent with those from the 2500 m analysis. Air parcels in summer more frequently originate from lower latitudes, while in winter, they primarily come from higher latitudes. We have included these boundary layer results alongside the original analysis to further support our discussion (Figure 6a in revised manuscript).

Additionally, we plotted the frequency distribution of the vertical heights of the back trajectories which demonstrate the subsidence in large scale. However, it is important to note

that ERA5 does not explicitly resolve entrainment processes, as they occur at sub-grid scales. Despite this limitation, we have included the vertical height distribution plot in the supplementary materials (Figure A1) for reference.

Section 3: To improve the flow and readability of this section, it might be helpful to introduce the patterns of the clusters (Figure 3) before discussing their frequency (Figure 1).

We appreciate your suggestion to improve the flow and readability of Section 3. In response, we have relabelled Figure 3 as Figure 1 in the revised manuscript to present the cluster patterns earlier. Additionally, we combined the original Figures 1 and 2 to better illustrate the consistency of the observed seasonality in our clusters with the STR, as advised by another reviewer. We hope these changes will make the manuscript clearer and easier to follow.

**Minor comments:**

*• Line 64. In this paper, the word "pristine" was used multiple times here and elsewhere. The meaning of pristine needs to be clarified (e.g. in Hamilton et al., 2014). Here, it referred to the Southern Ocean as pristine, while in other places, it seemed to suggest that pristine means low aerosol concentration. Hamilton, D. S., Lee, L. A., Pringle, K. J., Reddington, C. L., Spracklen, D. V., & Carslaw, K. S. (2014). Occurrence of pristine aerosol environments on a polluted planet. Proceedings of the National Academy of Sciences, 111(52), 18466–18471.* We have added a general note in line 31 to clarify this.

*• Line 130, Figure 1: Please consider adding the full names for each cluster in the caption.* The full names of each cluster have been added to the caption of the revised Figure 1 for clarity, as requested.

*• Line 133, Figure 2: The circles in the figure are not labeled in the legend and are not mentioned in the main text.* They are the outliers. We added this to the new figure 2 caption.

*• Line 204 & Line 196. Why was a different test used for mean precipitation (Student's t-test), while the Whitney U test was used for CCN?*
Due to the nature of precipitation over our study area as discussed before, the median precipitation is 0, which makes the mean more appropriate for capturing the range of precipitation events. In contrast, the median was used for CCN data to mitigate the influence of outliers (this has been clarified in lines 113-116). Subsequently, we used the two-tailed Student's t-test for precipitation as it is appropriate for comparing means between two groups. For CCN and radon in contrast, the Mann-Whitney U test was used as it is more suitable for medians. I have added a note in this regard in lines 226 and 212 respectively.

*• Line 230. What does it mean by "combine the two baseline clusters together"? Please clarify in the text.* Done (line 256).

*• Line 234 & Figure 6: To make understanding the orders of magnitude of rain rate more intuitive, please consider using a base-10 logarithmic scale (log10) for the precipitation rate instead of a natural logarithmic scale.* Done.
*• Line 259 & Figure 7: It might be helpful to show the back trajectories for S-base and W-base alongside their differences.* We already have shown the differences in their back trajectories at 500 m in figure 4.

Sincerely,

Tahereh Alinejadtabrizi

---

## Author Response (AR3)

School of Earth, Atmosphere and Environment,

Monash University, VIC 3800, Australia

Prof. Greg McFarquhar,

Journal of Atmospheric Chemistry and Physics

January, 2025

Re:

Once more, we sincerely appreciate the insightful feedback from both reviewers, which has greatly contributed to the improvement of our manuscript. Their valuable suggestions have significantly enhanced the clarity of our work.

In response to the technical comment regarding the description of Figure 5a, I carefully reviewed the relevant sentence. Previously, the manuscript incorrectly stated that the minimum CCN concentration occurs in summer, with a peak in winter. This has been revised to accurately reflect the data shown in Figure 5a, where the minimum $N_{CCN}$ occurs in winter, and it peaks in summer.

Additionally, I corrected a typographical error by correcting "kannaook" to "Kannaook" to ensure accuracy and improve the overall appearance of the manuscript.

Thank you again for your valuable input and for recommending acceptance.

**Reviewer 1.**

*Thanks for the updated version and I think this has largely improved. I would recommend accepting after confirming with the authors about a minor detail. For the line 268, it is mentioned that "...NCCN... (Figure 5a) with a peak concentration observed during winter and a minimum over summer". Please confirm the description is consistent with the Figure 5a before publishing.*

> We greatly appreciate the reviewer's valuable feedback provided during the previous rounds of revisions, which has truly enhanced the manuscript.
>
> In response to your technical comment, I have carefully reviewed the description of Figure 5a and revised the sentence in line 264 to ensure consistency. Thank you for bringing this to our attention—it was indeed an oversight on our part.

Sincerely,

Tahereh Alinejadtabrizi